# SILC: Improving Vision Language Pretraining with Self-Distillation

## Abstract

Image-Text pretraining on web-scale image caption dataset has become the default recipe for open vocabulary classification and retrieval models thanks to the success of CLIP and its variants. Several works have also used CLIP features for dense prediction tasks and have shown the emergence of open-set abilities. However, the contrastive objective only focuses on image-text alignment and does not incentivise image feature learning for dense prediction tasks. In this work, we propose the simple addition of local-to-global correspondence learning by self-distillation as an additional objective for contrastive pre-training to propose SILC . We show that distilling local image features from an exponential moving average (EMA) teacher model significantly improves model performance on several computer vision tasks including classification, retrieval, and especially segmentation. We further show that SILC scales better with the same training duration compared to the baselines. Our model SILC sets a new state of the art for zero-shot classification, few shot classification, image and text retrieval, zero-shot segmentation, and open vocabulary segmentation.

## 1 Introduction.

Recent advancements in self-supervised learning (Caron et al., 2021; Oquab et al., 2023; Chen et al., 2020; Grill et al., 2020) and weakly supervised learning on web data (Radford et al., 2021; Jia et al., 2021; Zhai et al., 2023) has spearheaded the development of foundational language(Radford et al., 2018; Chowdhery et al., 2022) and vision-language models (Radford et al., 2021; Jia et al., 2021; Zhai et al., 2023). These methods get around the long term challenge of obtaining large labelled dataset by developing self-supervision criterions. The development of Transformers (Vaswani et al., 2017; Dosovitskiy et al., 2021) has further facilitated this trend as Transformers scale better with larger datasets, e.g. such as the ones available from the internet.

Developing open vocabulary computer vision models that can reason beyond a pre-determined set of classes has been a long-term challenge. The introduction of web image-text datasets and the progress in compute have enabled significant advances in this field. Popularized by CLIP (Radford et al., 2021), contrastive pretraining utilizes large datasets with paired image and text from the web and trains a vision-language model (VLM) to embed them to a shared latent space. Since these models are trained on a wide set of concepts, the learned VLM allows for open vocabulary inference (Radford et al., 2021). However, developing open vocabulary dense prediction models for segmentation and detection is still an open challenge, since internet-scale dataset do not have labels available for these tasks. Several works have found that incorporating VLMs in segmentation and detection models can unlock some open vocabulary abilities (Cho et al., 2023; Ding et al., 2022; Xu et al., 2022b). Since CLIP is not trained for these tasks, these methods get around its limitations by tuning the learned model with some dense prediction labelled dataset. One set of methods utilizes a normal segmentation / detection model for class agnostic inference and then predict the class logits with CLIP (Cho et al., 2023; Liang et al., 2023). Another family of methods aims to distill VLMs directly into a dense prediction model and utilize the text transformer to generate the class weights to predict logits Li et al. (2022b); Ghiasi et al. (2022). These works have been highly impactful towards expanding open vocabulary abilities of dense prediction models. However, since the contrastive pretraining objective does not explicitly encourage learning good local features for dense prediction tasks, these methods are limited by the VLM's intrinsic performance (Oquab et al., 2023) as we also show later in our experiments.

In the self-supervised literature, enforcing local-to-global consistency by self-distillation has emerged as a powerful pretraining objective (Caron et al., 2021; Oquab et al., 2023; Zhou et al., 2022b) to learn vision backbones that are competitive on classification as well as dense prediction tasks, e.g. segmentation and detection. However, these backbones can not directly be used for zero-shot or open vocabulary inference. We take motivation from these two branches of literature and unify image-text contrastive pretraining and local-to-global consistency learning in one pretraining framework to propose our novel method SILC. SILC utilises web image-text dataset to learn one model that improves VLM performance on existing classification and retrieval tasks while especially improving performance on zero-shot and open vocabulary segmentation.

Our contributions are as follows: 1. We propose a novel training framework for VLMs that pairs contrastive pretraining on image-text data with self-distillation on web images. 2. We show that SILC scales better with training duration than baselines and achieves significant improvements on all VLM tasks. 3. We show that our learned model especially improves zero-shot segmentation and open vocabulary segmentation tasks. 4. We contribute a new foundation model that sets a new state of the art on zero-shot classification, few-shot classification, image-to-text and text-to-image retrieval, zero-shot semantic segmentation and open vocabulary semantic segmentation.

## 2 RELATED WORKS.

**Image-Text Pretraining.** Vision-language model (VLM) pretraining (Radford et al., 2021; Jia et al., 2021; Li et al., 2022c; Chen et al., 2023) aims to learn generic multimodal representations that generalize to a wide range of downstream tasks. Substantial progress has been made in this field recently towards better pretraining objectives (Jia et al., 2021; Wang et al., 2022b) and better large-scale image-text dataset (Radford et al., 2021; Chen et al., 2023). One of the most popular objective functions is contrastive learning (Radford et al., 2021; Jia et al., 2021) that pulls positive image and text pairs close and pushes negative ones apart in the joint embedding space. It is capable of scaling to a large-scale pretraining dataset and learning highly discriminative image and text features. Many works (Li et al., 2021b; Zhai et al., 2022b; 2023; Yao et al., 2021) in this direction have demonstrated improvements across zero-shot image classification and retrieval benchmarks.

Another line of research focuses on generative learning via autoregressive text generation (Wang et al., 2022b;a; Tschannen et al., 2023). Compared to the contrastive learning, generative learning usually performs better on text generation tasks e.g., image captioning and VQA. Finally, there are hybrid methods (Alayrac et al., 2022; Li et al., 2021a; Singh et al., 2022; Lu et al., 2019; Yu et al., 2022; Li et al., 2022c) that combine multiple objective functions including generative, contrastive and multi-task losses. While many VLMs (Radford et al., 2021; Wang et al., 2022b) mainly focus on learning global image-text alignment that benefits image-level downstream tasks, our work aims to develop a new VLM that benefits both image-level and pixel-level tasks. There have been a few attempts (Dou et al., 2022; Luo et al., 2023; Zhong et al., 2022; Dong et al., 2023) to improve VLMs for dense prediction tasks including object detection and semantic segmentation. However, they are either modeling the fine-grained patch-text interactions that are not scalable (Dou et al., 2022; Luo et al., 2023) or rely on additional bounding box annotations (Zhong et al., 2022; Li et al., 2022d). In this work we propose to pair image-text contrastive learning with self-distillation to learn a VLM.

**Self-supervised Learning.** Self-supervised learning is another popular pretraining paradigm where features are learned from image data itself. One branch of methods optimize the network to solve pretext tasks e.g., image coloring (Zhang et al., 2016), inpainting (Pathak et al., 2016), transformation prediction (Gidaris et al., 2018), and patch ordering (Misra & Maaten, 2020). Another family of approaches adopt instance-level discriminative learning via contrastive learning (Chen et al., 2020; He et al., 2020) and clustering (Caron et al., 2018; 2020). Recently, (He et al., 2022) shows that masked autoencoder is also a scalable self-supervised learner. Our work is inspired by DINO (Caron et al., 2021) which shows that segmentation emerges from learning local and global-views consistency. However, DINO cannot be directly used for zero-shot and open-vocabulary inference because it only learns image features. In contrast, our method is trained on image and text data jointly to enable more vision-language applications.

**Zero-shot Semantic Segmentation.** Zero-shot semantic segmentation aims to segment arbitrary visual concepts in the wild without dense annotations (Xu et al., 2022a). Methods in this area rely

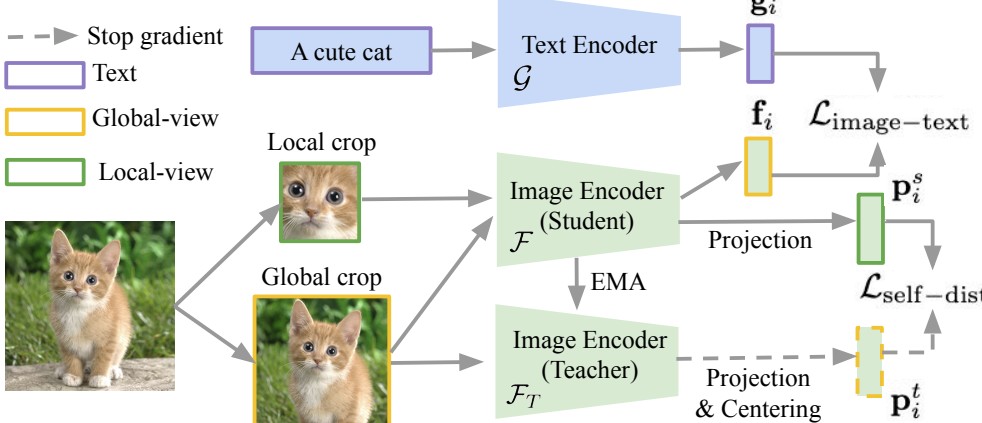

Figure 1: **SILC** is a two-tower transformer based VLM. The first component of our training objective uses a global view of an image covering a large area and its paired caption to optimise a batch-wise contrastive loss for images and texts. The second component of our training objective enforces local-to-global consistency by self-distillation between the main model (the student) and an Exponential Moving Average (EMA)-based teacher. This local-to-global correspondence additionally allows the model to learn good visual features. Together the two objectives allow the model to excel at both traditional VLM tasks as well as segmentation.

on image-text pairs from a combination of image captioning and web image-text dataset. Since these images do not have dense labels, these methods devise a self-supervised image region to text attention criterion. Group-VIT (Xu et al., 2022a) proposes to introduce grouping tokens that cluster similar image patches under each group token. MaskCLIP (Zhou et al., 2022a) found that normal CLIP training results in zero-shot segmentation emerging in the last transformer block of the image encoder. ReCo (Shin et al., 2022) proposes a refinement process on top of MaskCLIP by retrieval and co-segmentation. Finally the most recent state-of-the-art TCL (Cha et al., 2023) learns a decoder to upsample the grounded patch embeddings (values of last block) and learns a region to text attention.

**Open Vocabulary Segmentation.** Open-vocabulary semantic segmentation methods aim to segment images according to a vocabulary of class categories provided at test-time containing additional unseen classes. In contrast to zero-shot segmentation, open-vocabulary semantic segmentation has access to a semantic segmentation dataset with a limited vocabulary for training. Early methods for open-vocabulary semantic segmentation attempt to learn visual embeddings that align with existing text embeddings of the class names (Zhao et al., 2017; Xian et al., 2019; Bucher et al., 2019). With the emergence of large-scale vision-language pre-training such as CLIP, more recent methods transfer the open-vocabulary capabilities of CLIP from image- to pixel-level predictions. To achieve dense predictions, LSeg (Li et al., 2022a) learns pixel-wise visual embeddings that align with CLIP text embeddings while OpenSeg (Ghiasi et al., 2022) learns class-agnostic segmentation proposals to pool visual features for region-text grounding. To better preserve the zero-shot abilities of a pre-trained CLIP, ZegFormer (Ding et al., 2022) and ZSseg (Xu et al., 2022b) introduce a two-stage framework, which first learns class-agnostic segmentation mask predictions and classifies the corresponding region crops using a frozen CLIP. OVSeg (Liang et al., 2023) further finetunes CLIP on region-text pairs to compensate for the appearance shift of masked crops. To avoid the overhead of two stages, CAT-Seg (Cho et al., 2023) learns the aggregation of cost volumes between text embeddings and dense image embeddings from CLIP.

# 3 METHOD.

SILC builds on the contrastive pretraining framework of CLIP and consists of a two-tower transformer model with a shared embedding space. We utilize a web-scale paired image-text dataset and rely on large-scale pretraining to learn the weights of the model. The first component of our pretraining objective focuses on aligning matching image-text pairs close together and away from

other images and texts in the batch. This objective has been incredibly successful in recent literature. However, the contrastive objective in its current form does not focus on capturing rich local image semantics necessary for dense prediction tasks like segmentation. Therefore, we propose to pair the contrastive pretraining objective with a local-to-global consistency objective that uses self-distillation as shown in Figure 1. **SILC** gets its name from the two training objectives consisting of **S**elf-Distillation from Images and **I**mage-**L**anguage **C**ontrastive Alignment from Image-Text pairs.

### 3.1 ALIGNING IMAGE AND TEXT.

The contrastive pretraining objective relies on the Info-NCE framework (Oord et al., 2018). It utilizes large amount of web-scale image-text dataset to learn an alignment between paired image and text. Given a minibatch $\mathcal{B} = \{(I_1, T_1), (I_2, T_2), \dots\}$, where $(I_i, T_i)$ denotes a matching pairs of Image and Text, the contrastive objective encourages matching image and text pairs to lie close together in a shared embedding space. The image $I_i$ is processed by a learnable Vision Transformer $\mathcal{F}$ to get its feature embedding. Similarly, the tokenized text $T_i$ is processed by a learnable Text Transformer $\mathcal{G}$ to get its feature embedding. These feature embeddings are normalized by their $l_2$ norm to get $\mathbf{f}_i = \frac{\mathcal{F}(I_i)}{\|\mathcal{F}(I_i)\|_2} \in \mathbb{R}^J$ for the image $I_i$ and $\mathbf{g}_i = \frac{\mathcal{G}(T_i)}{\|\mathcal{G}(T_i)\|_2} \in \mathbb{R}^J$ for the paired text $T_i$ where $J$ is the feature dimension of the shared embedding space. The dot product of $\mathbf{f}_i$ and $\mathbf{g}_i$ computes their cosine similarity and is optimized with a pair of cross-entropy losses as follows:

$$\mathcal{L}_{\text{image-text}} = -\frac{1}{2|\mathcal{B}|} \sum_{i=1}^{|\mathcal{B}|} \left( \overbrace{\log \frac{e^{t\mathbf{f}_i \cdot \mathbf{g}_i}}{\sum_{j=1}^{|\mathcal{B}|} e^{t\mathbf{f}_i \cdot \mathbf{g}_j}}}^{\text{image} \to \text{text softmax}} + \overbrace{\log \frac{e^{t\mathbf{f}_i \cdot \mathbf{g}_i}}{\sum_{j=1}^{|\mathcal{B}|} e^{t\mathbf{f}_j \cdot \mathbf{g}_i}}}^{\text{text} \to \text{image softmax}} \right), \tag{1}$$

where $t$ is the learnable temperature and controls the peakness of the activation in the loss function. The batch-wise contrastive loss relies on a large batch size to align image-text pairs. This objective tuned over a large amount of data learns a shared embedding space between image and text and thus can be used for zero-shot transfer to several computer vision tasks like classification and retrieval.

### 3.2 DISTILLING LOCAL IMAGE FEATURES.

The image-text contrastive loss has shown to be very successful in learning zero-shot transfer models (Radford et al., 2021; Jia et al., 2021). Models learned with this objective have also been used to improve dense prediction tasks like open vocabulary segmentation and detection. However, the contrastive objective alone does not explicitly focus on learning good visual features for dense prediction tasks. These tasks require local image semantics to be sufficiently encoded in the output image and patch embeddings. Enforcing local-to-global consistency has emerged as a powerful technique to accomplish this on large unlabelled image data (Caron et al., 2021; Oquab et al., 2023; Zhou et al., 2022b) in self-supervision literature. However, these methods can not be directly used for open vocabulary models. In the second component of our training framework, we take inspiration from this subset of literature and additionally add local-to-global consistency as a training objective for images in our image-text dataset.

The basic idea of this objective is as follows. A teacher network gets a global view of the image representing the scene as a whole and produces a feature embedding. A student model gets a partial view of the same image and produces a feature embedding. A self-distillation objective is introduced where the student needs to match the prediction of the teacher while only having partial information of the scene. This enforces the model to learn local semantics and their relation to global semantics of the scene. We add this criterion for the image encoder $\mathcal{F}$. We add a projection as a learnable MLP on top of image encoder to map from the original shared embedding space of dimension $J$ to $K$ where $K > J$. The student $\mathcal{F}_S$ is the main image encoder with a learnable projection head. Since we rely on noisy web scale image-text data, we do not have an oracle teacher for the student to match. We therefore construct our teacher $\mathcal{F}_T$ as a exponential moving average of the student $\mathcal{F}_S$ from the previous training iterations to realize our self-distillation framework:

$$\mathcal{F}_T \leftarrow \lambda \mathcal{F}_T + (1 - \lambda)\mathcal{F}_S, \tag{2}$$

where $\lambda$ controls the update step of the teacher. For a given image $I_i$, the teacher processes its global crop to produce $\mathbf{p}_i^t \in \mathbb{R}^K$ and the student processes its local crop to produce $\mathbf{p}_i^s \in \mathbb{R}^K$.

| Model | Zero-Shot Classification | | Few-shot classification | | | | | | Retrieval | |
| | Imagenet | CIFAR100 | Imagenet | | | CIFAR100 | | | Coco | |
| | T1 | T1 | 1shot | 5shot | 10shot | 1shot | 5shot | 10shot | I2T@1 | T2I@1 |
|---|---|---|---|---|---|---|---|---|---|---|
| SLIP (Mu et al., 2022) | 73.8 | 69.3 | 42.3 | 62.7 | 66.9 | 40.8 | 61.0 | 65.8 | 61.6 | 44.3 |
| XCLIP (Zhou et al., 2023) | 74.1 | 68.5 | 44.1 | 62.5 | 66.1 | 36.2 | 58.8 | 63.7 | 60.6 | 41.8 |
| SigLIP (Zhai et al., 2023) | 75.1 | 69.8 | 44.0 | 64.2 | 68.4 | 39.0 | 61.7 | 66.3 | 62.6 | 44.9 |
| MaskCLIP (Dong et al., 2023) | 74.4 | 69.0 | 42.6 | 61.0 | 64.7 | 44.1 | 63.2 | 67.6 | 61.4 | 43.6 |
| CLIP (WebLI) (Zhai et al., 2023) | 74.1 | 68.4 | 42.8 | 63.2 | 67.3 | 39.4 | 59.6 | 64.6 | 61.7 | 43.9 |
| **SILC\* (Ours)** | 75.3 | 71.0 | 44.6 | 64.3 | 67.8 | 42.8 | 64.6 | 69.6 | 62.5 | 44.9 |
| **SILC (Ours)** | **76.2** | **72.3** | **45.3** | **65.0** | **68.5** | **45.2** | **66.9** | **71.3** | **66.1** | **49.1** |

Table 1: **Comparing SILC\* with baseline**, we observe that our pretraining framework results in significant improvement over various VLM pretraining methods. We reproduce all methods on the same WebLI dataset (Chen et al., 2023) to quantify the improvements from the training objective. We further fine-tune SILC\* on a cleaner subset to get our final model SILC and see that it unlocks additional performance without significant extra retraining. The best performance at each model size is **bolded**, the second best is underlined.

To prevent the teacher from collapsing to a trivial solution, we apply sharpening on the outputs of teacher with $\tau_t$ and student with $\tau_s$. To encourage each feature dimension to contribute to the output feature, we additionally introduce a centering operation on the prediction of the teacher. The centering term $\mathbf{c} \in \mathbb{R}^K$ is initialized with 0 and is updated by a momentum update with a factor of $m$ with the first order batch statistics of the teacher's prediction at each step as follows: $\mathbf{c} \leftarrow m\mathbf{c} + (1-m)\frac{1}{|\mathcal{B}|}\sum_{i=1}^{|\mathcal{B}|}\mathbf{p}_i^t$.

To learn local-to-global correspondences, the student is faced with an information asymmetry. The student is given a local view of an image which is realized as a random crop over a small region of the image. The teacher, however, has access to a global view of the image containing more information about the scene. The student is tasked with matching the semantics of the teacher while only having partial information. Therefore, for a given image, the model needs to learn local semantics of the image and how it would fit in the global context of this image. This is realized as a knowledge-distillation loss where the student and the teacher's feature vectors are first converted to a probability distribution by applying a softmax on the teacher prediction $\mathcal{P}_t(I_i^{gl}) = \texttt{softmax}((\mathbf{p}_i^t - \mathbf{c})/\tau_t)$ and student prediction $\mathcal{P}_s(I_i^{lc}) = \texttt{softmax}(\mathbf{p}_i^s/\tau_s)$. The student is optimized to match the teacher's output with a cross-entropy loss.

$$\mathcal{L}_{\text{self}-\text{dist}} = -\mathcal{P}_t(I_i^{gl})^\intercal log(\mathcal{P}_s(I_i^{lc})) \tag{3}$$

This self-distillation objective incentivises the image encoder to learn local semantics of images over the large web scale dataset. Since the teacher is constructed with the student's weights, and the image level features are pooled from patch embeddings in a Vision Transformer, this allows for richer local semantics to be captured in the image level as well as the patch level features.

## 4 EXPERIMENTS.

We compare SILC with several image-text pretraining methods on the same test bench and perform extensive experimentation. We show that SILC sets a new state of the art on a variety of tasks: zero-shot classification, few-shot classification, retrieval, zero-shot segmentation and open vocabulary segmentation.

### 4.1 IMPLEMENTATION DETAILS.

We implement our model in jax in the `big_vision` codebase (Beyer et al., 2022a;b), following the contrastive pre-training setups from (Zhai et al., 2023), and use the WebLI dataset(Chen et al., 2023) for our experiments. We utilize two global views cropped between (0.4-1.0) of the original image area and eight local views cropped between (0.05-0.4) of the original image area for the self-distillation loss. The global views are resized to $(256 \times 256)$ and the local views are resized to $(96 \times 96)$. The teacher momentum $\lambda$ is kept fixed at 0.966 and the center update momentum $m$ is kept fixed at 0.9 through the training. The teacher temperature $\tau_t$ is fixed at 0.04 and the student temperature $\tau_s$ is fixed at 0.1. $K$ is 65536. We resize the original image to $(256 \times 256)$

for the contrastive loss between image-text pairs. We additionally use each augmented global view for the contrastive loss with the text. However the batch size for negatives is kept the same. We trained with a batch size of 16k on Google TPUs. We use *example-seen* to represent how many image and text pairs are drawn from the dataset throughout the training. We train all baselines in our main comparisons in Table 1 for 20 Billion example-seen on the WebLI dataset (Chen et al., 2023) following Zhai et al. (2023). Our model trained on WebLI is marked as **SILC\*** . We use a rsqrt learning scheduler Zhai et al. (2022a) with base learning rate of 0.001 with 50000 warm up and 50000 cool down steps. We additionally finetune our model using a smaller but cleaner WebLI subset (Chen et al., 2023) for 1 Billion additional example-seen and represent this model as **SILC** . The smaller WebLI subset contains 100 million image-text pairs with finer-grained text filters etc.

## 4.2 State-of-the-art comparison on Classification and Retrieval.

**Compared Baselines.** We compare with several popular image-text pretraining methods under the same training and evaluation protocol in Table 1. CLIP Radford et al. (2021) is the most popular baseline trained with the contrastive loss introduced in Section 3.1. We refer to CLIP trained on WebLI as CLIP (WebLI) to avoid confusion with other popular variants trained on other datasets. SLIP (Mu et al., 2022) proposes to incorporate Sim-CLR loss between two augmented global view of the image in addition to the CLIP loss on the resized image. xCLIP (Zhou et al., 2023) proposes to add a self-distillation loss between the features of the text-transformer and the image in addition to the CLIP loss. MaskCLIP (Dong et al., 2023) proposes to use the original image and an additional masked image in the contrastive loss and self-distillation. SigLIP (Zhai et al., 2023) proposes to replace the contrastive loss in CLIP with the Sigmoid loss. SLIP, XCLIP and MaskCLIP only train for short durations with limited data in their respective manuscripts. For fair comparison, we train all baselines using the same WebLI dataset following setups described in (Zhai et al., 2023).

We compare with these baseline VLMs in Table 1at ViT/B16 and see that our model consistently achieves state-of-the-art performance on all metrics. SILC\* also consistently improves on CLIP for all metrics. On zero-shot classification on Imagenet, SILC\* improves on CLIP (WebLI) by 1.2 points, similarly we notice an improvement of 2.6 points on CIFAR-100 showing the benefit of local feature self-distillation. Similar improvements are noted for few-shot classification where SILC\* improves over CLIP (WebLI) by 1.8, 1.1 and 0.5 points on ImageNet 1 shot, 5 shot and 10 shot classification respectively. We notice similar improvements on retrieval where SILC\* shows improvements on image to text as well as text to image retrieval. We observe that SLIP achieves very similar performance to CLIP (WebLI). We noticed similar improvement to the author's work in early stage of training but when trained for longer, SLIP converged to the same point as CLIP (WebLI). This shows that the Sim-CLR loss' contribution to model performance diminishes when training CLIP at scale. As we compare with XCLIP, we again notice that this baseline performs very similarly to CLIP (WebLI) when trained for 20B example-seen. Comparing MaskCLIP Dong et al. (2023), we observe that this baseline adds some performance for zero-shot classification over CLIP (WebLI) but behaves very similarly on other metrics. Finally, we see that SigLIP improves on CLIP (WebLI) performance on all metrics but SILC\* continuously outperforms it on most metrics indicating that our training framework is the state-of-the art in zero-shot classification, few-shot classification and retrieval compared to baseline works.

Comparing SILC\* with SILC , we notice that the finetuning on cleaner subset unlocks additional performance for the model without significant extra training. We notice another 0.9 point improvement over SILC\* on zero-shot ImageNet classification with SILC . We observe improvements of the same magnitude on few-shot classification. Comparing retrieval performance, we see a significant increase in retrieval performance on Coco where SILC achieves a 3.6 and 4.2 points improvement on Image to Text and Text to Image Recall@1.

## 4.3 Zero-Shot Semantic Segmentation.

Zero-shot semantic segmentation aims to measure the grounding performance of a VLM usually from its patch embeddings. MaskCLIP (Zhou et al., 2022a) (different from MaskCLIP Dong et al. (2023)) found that for the original CLIP model, this grounding emerges in the values of the last transformer encoder block's MHA. We use a Vision Transformer with a MAP pooling head (Zhai et al., 2022a). We observe that grounding for our model emerges in the values of the MAP head

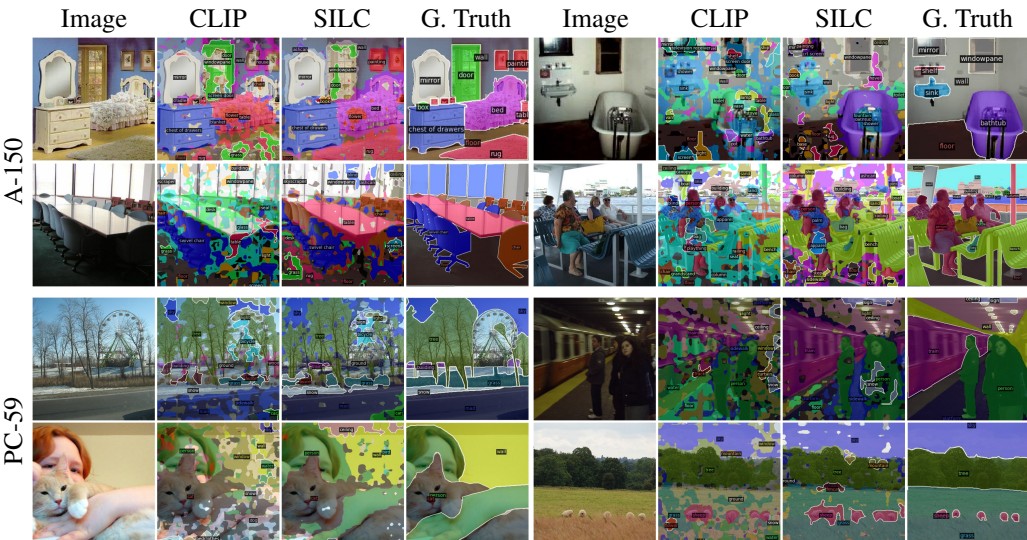

Figure 2: **Qualitative results on zero-shot segmentation** show that SILC achieves significant improvements over CLIP (WebLI). SILC produces less noisy segmentation and better distinguishes semantic classes. This semantic segmentation emerges without any segmentation supervision.

| Model | A-150 | PC-59 | Cityscapes | VOC-20 | COCO-Stuff |
|---|---|---|---|---|---|
| GroupVIT (Xu et al., 2022a) | 9.2 | 23.4 | 11.1 | **79.7** | 11.1 |
| MaskCLIP (Zhou et al., 2022a) | 9.8 | 26.4 | 12.6 | 74.9 | 16.4 |
| ReCo (Shin et al., 2022) | 11.2 | 22.3 | 21.1 | 57.7 | 14.8 |
| TCL (Cha et al., 2023) | 14.9 | 30.3 | 23.1 | 77.5 | 19.6 |
| CLIP (WebLI) (Zhai et al., 2023) | 15.0 | 24.0 | 22.6 | 69.5 | 15.0 |
| **SILC* (Ours)** | 17.2 | 29.3 | 25.1 | 73.5 | 18.2 |
| **SILC (Ours)** | **19.3** | **31.6** | **26.9** | 77.5 | **20.8** |

Table 2: **Comparing Zero-Shot Segmentation performance** we see that SILC* trained on noisy web image-text data already outperforms several ZS segmentation baselines that use cleaner image-text data. When we tune our model on a cleaner subset of image-text data to get SILC , we see that it sets the absolute state-of-the-art on 4/5 datasets. We would also like to emphasize that SILC achieves this without learning an expensive image patch to text attention that TCL relies on.

instead of the last encoder block. These values are processed by the layer norm and MLP layers of the MAP pooling head to get output patch embeddings. For a given set of possible classes in a segmentation dataset, we obtain the corresponding text embeddings by querying our text encoder with a standard prompt. We compute the cosine similarity between the image patch embeddings and the text features of each class name to generate a segmentation map in zero-shot. We report the mean-IOU (mIOU) performance of our model in Table 2 and compare with baselines at ViT/B16 similar to previous works. We follow the evaluation protocol of TCL (Cha et al., 2023) without the background class. However, we do not use any post-refinement step e.g. PAMR as we argue that the raw segmentation of a VLM is the true depiction of its zero-shot segmentation performance.

As we compare SILC* with CLIP (WebLI), we see that our knowledge distillation setup for local-to-global correspondences improves the zero-shot segmentation performance consistently by a few mIOU points on all 5 datasets. In fact SILC* is also superior to ReCo and GroupVIT while being competitive to TCL without learning an expensive image patch to text attention. SILC* is trained on noisier web dataset compared to GroupVIT (Xu et al., 2022a), ReCo (Shin et al., 2022) and TCL (Cha et al., 2023) which use relatively cleaner smaller image caption datasets. When we fine-tune SILC* with cleaner subset of data to get SILC , we notice a significant improvement on all datasets. Compared to the previous state-of-the-art TCL, SILC achieves a remarkable 4.3 mIOU points improvement on A-150, 2.9 points improvement on PC-59, and 4.9 points improvement on

| VLM | Method | A-847 | PC-459 | A-150 | PC-59 | VOC-20 | VOC-21 |
|---|---|---|---|---|---|---|---|
| CLIP-B/16 | ZegFormer (Ding et al., 2022) | 5.6 | 10.4 | 18.0 | 45.5 | 89.5 | 65.5 |
| CLIP-B/16 | ZSseg (Xu et al., 2022b) | 7.0 | - | 20.5 | 47.7 | 88.4 | - |
| CLIP-B/16 | OVSeg (Liang et al., 2023) | 7.1 | 11.0 | 24.8 | 53.3 | 92.6 | - |
| CLIP-B/16 | CAT-Seg (Cho et al., 2023) | 8.4 | 16.6 | 27.2 | 57.5 | 93.7 | 78.3 |
| **SILC-B/16** | CAT-Seg (Cho et al., 2023) | 13.4 (+5.0) | 22.0 (+5.4) | 36.6 (+9.4) | 61.2 (+3.7) | 95.9 (+2.2) | 80.4 (+2.1) |
| CLIP-L/14 | ZSseg (Xu et al., 2022b) | 7.1 | 10.2 | 21.7 | 52.2 | 92.3 | - |
| CLIP-L/14 | OVSeg (Liang et al., 2023) | 9.0 | 12.4 | 29.6 | 55.7 | 94.5 | - |
| CLIP-L/14 | CAT-Seg (Cho et al., 2023) | 10.8 | 20.4 | 31.5 | 62.0 | 96.6 | 81.8 |
| **SILC-L/16** | CAT-Seg (Cho et al., 2023) | 15.0 (+4.2) | 25.8 (+5.4) | 37.7 (+6.2) | 63.5 (+1.5) | 97.6 (+1.0) | 82.5 (+0.7) |
| CLIP-G/14 | CAT-Seg (Cho et al., 2023) | 13.3 | 21.4 | 36.2 | 61.5 | 97.1 | 81.4 |

Table 3: **Comparing Open Vocabulary Semantic Segmentation performance**, we observe that SILC significantly improves over CLIP by significant margins on all unseen test sets. SILC particularly improves the performance for challenging test sets with large vocabularies. SILC-L/16 even outperforms the much larger CLIP-G/14. All models are trained on COCO-Stuff.

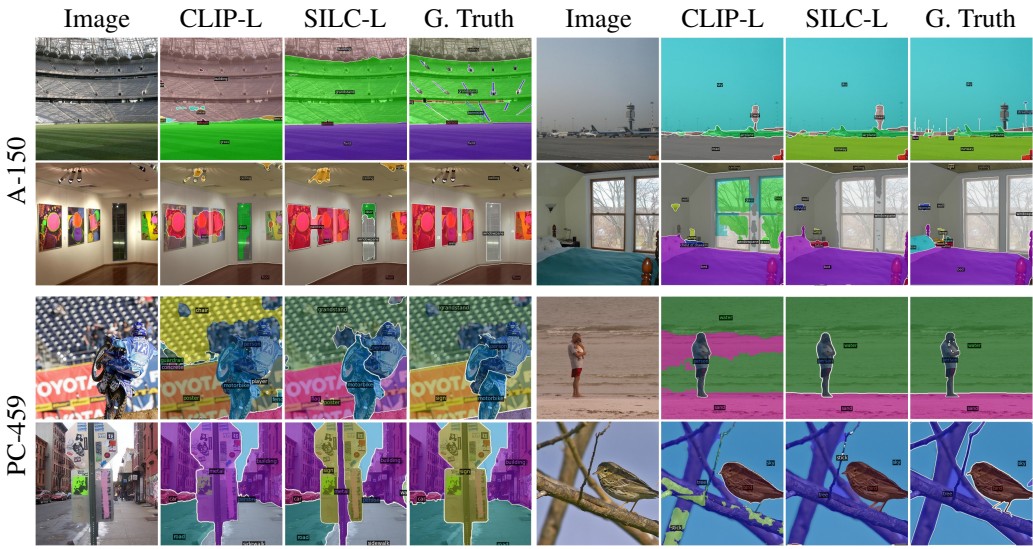

Figure 3: **Comparing qualitative examples for open vocabulary segmentation**, we observe that SILC w/ CAT-Seg better distinguishes semantically similar classes such as field/grass, runway/road, grandstand/chair, sand/water, sign/metal, and tree/stick than CLIP.

CityScapes. Similar improvements are noted on VOC-20 and COCO-Stuff however Group-VIT maintains the best result on VOC-20. In our preliminary experiments we noticed that the improvements in zero-shot segmentation are achievable by finetuning on a cleaner subset of data. We did not observe superior performance by learning an expensive patch-wise attention similar to TCL. We show the improvements of SILC on CLIP (WebLI) qualitatively in Figure 2. We can observe that SILC is better at segmenting and labeling semantic classes in images. We would like to emphasize that SILC has achieved this this without any segmentation ground truth.

## 4.4  OPEN-VOCABULARY SEMANTIC SEGMENTATION.

Open Vocabulary Semantic Segmentation aims to develop segmentation models that can segment novel classes beyond the training vocabulary. Most of the recent methods in this area rely on a pretrained CLIP due to its open-vocabulary capabilities and adapt it for segmentation task. To evaluate the open vocabulary segmentation potential of SILC , we take the current state-of-the-art model CAT-Seg (Cho et al., 2023) and replace the CLIP model used by the authors with SILC . The models are trained on COCO-Stuff-164k with 172 classes and tested on unseen datasets with different vocabularies: ADE-20k with 847 or 150 classes (A-847/A-150), Pascal Context (PC-459/PC-59), and Pascal VOC (VOC-20/VOC-21).

| Model | Imagenet 0 shot | Imagenet Few shot | | | Coco Retrieval | | ZS Segmentation | | | Open Vocab Seg | | |
|---|---|---|---|---|---|---|---|---|---|---|---|---|
| | T1 | 1shot | 5shot | 10shot | I2T@1 | T2I@1 | A-150 | Stuff | PC-59 | PC-459 | A-150 | PC-59 |
| CLIP (WebLI) | 71.7 | 36.4 | 57.7 | 62.5 | 59.1 | 42.9 | 11.8 | 12.9 | 20.1 | 18.6 | 30.5 | 57.7 |
| + additional views | 73.6 | 38.7 | 60.8 | 65.7 | 60.6 | 43.2 | 11.7 | 13.0 | 20.0 | 19.2 | 32.1 | 57.8 |
| + EMA | 73.7 | 38.4 | 60.7 | 65.5 | 61.3 | 43.1 | 11.9 | 13.3 | 20.5 | 19.0 | 32.2 | 57.5 |
| + Self Dist (**SILC***) | **74.3** | **39.9** | **61.2** | **65.7** | **62.7** | **43.9** | **12.2** | **15.3** | **21.1** | **21.0** | **33.3** | **60.7** |

Table 4: **We ablate over each component** of our model to verify our design choices. The addition of additional image augmentation and EMA to CLIP (WebLI) improves classification and retrieval metrics but only slightly impact the segmentation. Adding local-to-global consistency by self-distillation, we observe an improvement across the board especially on segmentation metrics.

From Table 3, we observe that SILC significantly improves on CLIP. In fact, SILC -B/16 performs on par with the much bigger CLIP-G/14 on the three most challenging test datasets A-847, PC-459 and A-150. The observed improvements of SILC also transfer to the larger ViT-L variant, where CAT-Seg with SILC-L/16 outperforms CAT-Seg with CLIP-L/14 on all datasets by a significant margin. In particular, it achieves more than +4 mIOU improvement on the challenging A-847, PC-459, and A-150. SILC-L/16 even significantly outperforms the much bigger CLIP-G/14 on all tested datasets. The improvements of SILC over CLIP are also reflected in the qualitative examples in Fig. 3. We observe that SILC better distinguishes semantically similar classes such as grandstand/building, field/grass, runway/road, grandstand/chair, and sign/metal. Further, it improves segmentation in difficult cases (such as the paintings) and better handles transparent segments (such as windowpane).

### 4.5 ABLATION ON MODEL COMPONENTS.

We ablate on the various design choices of our model and their impact on various tasks. We train all models for 5 Billion example-seen and report the performance in Table 4. Since our method processes additional image augmentations in the contrastive loss, we first test if our improvements are a consequence of processing more augmentations. We observe that the introduction of additional image augmentations (second row) improve the classification and retrieval metrics but their impact on zero-shot segmentation and open vocabulary segmentation is not as significant. When we add an EMA over this model's weights similar to our model (third row), we notice a slight improvement for the EMA model as seen in previous SSL literature. Finally when we add the self-distillation from local crops, we see an improvement across the board on all tasks. This improvement is more profound on the segmentation tasks highlighting our proposal's impact on these tasks.

### 4.6 HOW DOES SILC* SCALE WITH EXAMPLE-SEEN?

We test CLIP (WebLI) trained with additional data augmentations in each forward pass in Section 4.5. We additionally test the examples-seen efficiency of SILC* and compare to CLIP (WebLI). We report the results in Table 5. We notice that SILC* at only 5B examples-seen already achieves superior performance to CLIP (WebLI). When we further train SILC* to 10B and 20B example, we notice even superior performance.

| | CLIP (WebLI) | SILC* | | |
|---|---|---|---|---|
| Example-Seen | 20B | 5B | 10B | 20B |
| IM 0 shot | 74.1 | 74.3 | 75.0 | 75.3 |

Table 5: SILC* shows greater efficiency for example seen.

## 5 CONCLUSION.

We propose to integrate local-to-global correspondance learning by self-distillation as a complementary objective to the popular VLM Contrastive objective originally proposed by CLIP (Radford et al., 2021). We show that the introduction of this allows the VLM to scale better to example-seen from the dataset and results in significant performance improvements on multiple computer vision tasks. We see a consistent performance improvement on zero-shot classification, few-shot classification, and retrieval. We further test our VLM on zero-shot segmentaton and show that our training framework results in significant improvements without using any dense ground truth. Finally we show that SILC can significantly improve the open vocabulary segmentation performance thanks to our training framework. SILC sets a new state-of-the-art in Vision-Language Foundational Models.

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
