# Supplementary for SILC: Improving Vision Language Pretraining with Self-Distillation

## 1 Detail about supplementary.

In this supplementary, we provide the following additional experiment and detail about our work.

- Section 2: Classification and Retrieval performance across different model size.
- Section 3: Performance of Teacher vs Student.
- Section 4: Additional Results on Open Vocabulary Segmentation.
- Section 5: Additional Results on Zero-Shot Semantic Segmentation.
- Section 6: Additional Qualitative Results.
- Section 7: Additional details about evaluation and training.

## 2 Classification and Retrieval performance of additional SILC models.

In our main manuscript Table 1, we show that SILC* and SILC consistently improve over other image-text pretraining methods at ViT/B16 size. We additionally train CLIP (WebLI), SILC* and SILC with ViT/L16 to show that our improvements are consistent at larger model size too. We also train SILC* and SILC at ViT/B8 to study the trade off between model size vs patch size. We report the results in Table 1. Comparing SILC* with CLIP at ViT/16, we observe that our model consistently improves over the baseline to set a new state-of-the-art at this model size too. SILC* achieves a 1.3 points improvement over CLIP (WebLI) on ImageNet zero-shot classification. Similar improvements are noted over other classification and retrieval metrics. Finetuning SILC* to get SILC shows consistent improvements over all metrics showing that the cleaner subset benefits the larger model too. We also train SILC* with ViT/B8. ViT/B8 has the same number of learnable parameters for the Transformer as B/16 but uses half the patch size. We observe that the smaller patch size allows this model to consistently outperform the B/16 model. However, the smaller patch-size also means that the transformer has to process a longer sequence of tokens at each encoder block. As a result, ViT/B8 has approximately the same compute requirement as ViT/L16. We observe that The B/8 model performs slightly worse than the L16 model. Finally we see that the B/8 model also benefits from finetuning on the cleaner subset of WebLI and SILC B/8 consistently improves on SILC* B/8.

## 3 Performance of Teacher vs Student for SILC* .

Our training setup consists of the student that is updated with gradient descent and a teacher that is updated with an EMA update. For comparisons in our main manuscript, we report the performance for the teacher. We additionally compare the teacher with the student in Table 2. During training we observe that the teacher converges faster than the student but both converge to about the same performance towards the end of training for zero-shot classification, few-shot classification and retrieval. However for zero-shot segmentation, the teacher achieves superior performance compared to the student. Similar observation has been made by earlier self-supervised works Caron et al. (2021); Oquab et al. (2023) for self-supervised models. However in their case, the teacher always outperforms the students. In our setup, since the student is updated with image-text loss, it achieves similar performance to the teacher on the tasks listed before.

| Model | | Zero-Shot Classification | | Few-shot classification | | | | | | Retrieval | |
|---|---|---|---|---|---|---|---|---|---|---|---|
| | | Imagenet | CIFAR100 | Imagenet | | | CIFAR100 | | | Coco | |
| | | T1 | T1 | 1shot | 5shot | 10shot | 1shot | 5shot | 10shot | I2T@1 | T2I@1 |
| SILC* | ViT/B8 | 77.5 | 72.6 | 48.9 | 67.3 | 70.7 | 47.9 | 68.6 | 73.1 | 64.5 | 46.0 |
| SILC | ViT/B8 | **78.2** | **73.2** | **49.5** | 67.8 | 71.1 | 49.3 | 69.7 | 73.8 | 67.3 | 50.3 |
| CLIP (WebLI) (Zhai et al., 2023) | ViT/B16 | 74.1 | 68.4 | 42.8 | 63.2 | 67.3 | 39.4 | 59.6 | 64.6 | 61.7 | 43.9 |
| **SILC\* (Ours)** | ViT/B16 | 75.3 | 71.0 | 44.6 | 64.3 | 67.8 | 42.8 | 64.6 | 69.6 | 62.5 | 44.9 |
| **SILC (Ours)** | ViT/B16 | **76.2** | **72.3** | **45.3** | 65.0 | 68.5 | 45.2 | 66.9 | 71.3 | 66.1 | 49.1 |
| CLIP (WebLI) (Zhai et al., 2023) | ViT/L16 | 79.7 | 77.5 | 52.9 | 72.1 | 75.5 | 42.6 | 69.3 | 73.7 | 67.7 | 48.9 |
| **SILC\* (Ours)** | ViT/L16 | 81.0 | 80.5 | 54.8 | 73.9 | 76.8 | 53.2 | 75.8 | 79.5 | 68.4 | 50.9 |
| **SILC (Ours)** | ViT/L16 | **81.4** | **81.4** | **55.6** | 74.2 | 76.9 | 53.7 | 77.2 | 80.5 | 70.1 | 52.8 |

Table 1: **Performance of additional SILC models.** We show that SILC* outperforms CLIP (We-bLI) at ViT/L16 too. Moreover, we show that SILC achieves consistent improvement over SILC* at ViT/B8, ViT/B16 and ViT/L16. Best number for each model configuration is **bolded**. Second best is underlined.

| Model | Imagenet 0 shot | CIFAR 0 shot | Imagenet Few shot | | | Coco Retrieval | | ZS Segmentation | | |
|---|---|---|---|---|---|---|---|---|---|---|
| | T1 | T1 | 1shot | 5shot | 10shot | I2T@1 | T2I@1 | A-150 | Stuff | PC-59 |
| SILC* Teacher | 75.3 | 71.0 | 44.6 | 64.3 | 67.8 | 62.5 | 44.9 | 17.2 | 18.2 | 29.3 |
| SILC* Student | 75.3 | 71.0 | 44.6 | 64.3 | 67.8 | 62.5 | 44.9 | 16.1 | 17.3 | 27.4 |

Table 2: **Comparing SILC\* Teacher and Student performance,** we observe that both teacher and student behave similarly on classification and retrieval tasks. However, the teacher achieves superior performance on zero-shot segmentation.

# 4 ADDITIONAL RESULTS FOR OPEN VOCABULARY SEGMENTATION.

We report the Cat-Seg (Cho et al., 2023) performance reported by the authors in our Table 3 of the main manuscript. We show that SILC consistently improves on CLIP for open vocabulary segmentation to set a new state-of-the-art. However, the Cat-Seg manuscript uses CLIP trained on different image-text dataset. Moreover, in Table 4 of the main manuscript, we compare Cat-Seg performance using VLM trained on the same image-text dataset to show consistent improvements. This VLM is trained for 5 Billion Example-Seen. In this supplementary, we additionally report the performance improvements against CLIP (WebLI) trained for full 20 Billion Example-Seen. We report the results in Table 3 and show that our improvements are consistent. SILC consistently outperforms CLIP (WebLI) on all datasets.

# 5 ADDITIONAL RESULTS FOR ZERO-SHOT SEMANTIC SEGMENTATION.

TCL (Cha et al., 2023) the previous state-of-the-art in zero-shot semantic segmentation ensembles their learned model with MaskCLIP (Zhou et al., 2022) by tuning a mixing factor on the predictions of the two models. However, this mixing factor violates the zero-shot protocol proposed by (Xian et al., 2018) as the model has access to segmentation labels during the mixing factor tuning. We additionally report the performance of TCL using author's checkpoint by removing the ensemble with MaskCLIP in Table 4. We show that this results in slight drop in performance. We advice future works to not touch segmentation labels to tune parts of their models to be consistent with the zero-shot protocol. The TCL (Cha et al., 2023) results reported by the authors in their main paper additionally use PAMR to refine the predicted segmentation of their model and remove some noise. The authors also report the performance of their model without PAMR in their supplementary which we have reported in our main manuscript. We also list TCL with PAMR numbers in Table 4 to show that post refinement can give boost in performance but it can mask the actual performance of the learned model. Refinement steps can improve all methods as shown in TCL's supplementary. Therefore, we do not use refinement in our work as we are interested in the raw zero-shot segmentation performance of the model.

We also report zero-shot semantic segmentation results on an additional baseline PACL (Mukhoti et al., 2023) in Table 4. Since PACL checkpoints and code are not available, we contacted the authors and closely followed their instructions in our implementation. We train a small MLP as a residual on top of our CLIP (WebLI) B/16 model. We use the cleaner small subset of WebLI with 100 Million

| VLM | Method | A-847 | PC-459 | A-150 | PC-59 | VOC-20 | VOC-21 |
|---|---|---|---|---|---|---|---|
| **SILC-B/8** | CAT-Seg (Cho et al., 2023) | 15.0 | 24.3 | 38.7 | 63.4 | 96.7 | 81.5 |
| CLIP-B/16 (WebLI) | CAT-Seg (Cho et al., 2023) | 11.8 | 19.5 | 33.9 | 59.0 | 93.7 | 78.7 |
| **SILC-B/16** | CAT-Seg (Cho et al., 2023) | 13.4 (+1.6) | 22.0 (+2.5) | 36.6 (+2.2) | 61.2 (+2.2) | 95.9 (+2.2) | 80.4 (+1.7) |
| CLIP-L/16 (WebLI) | CAT-Seg (Cho et al., 2023) | 13.8 | 22.6 | 36.8 | 61.6 | 95.7 | 80.1 |
| **SILC-L/16** | CAT-Seg (Cho et al., 2023) | 15.0 (+1.2) | 25.8 (+3.2) | 37.7 (+0.9) | 63.5 (+1.9) | 97.6 (+1.9) | 82.5 (+2.4) |

Table 3: **Comparing Open Vocabulary Semantic Segmentation performance** against CLIp (We-bLI), we observe that the improvements of SILC are consistent on both model sizes.

| Model | A-150 | PC-59 | Cityscapes | VOC-20 | COCO-Stuff |
|---|---|---|---|---|---|
| TCL + PAMR (Cha et al., 2023) | 17.1 | 33.9 | 24.0 | 83.2 | 22.1 |
| PACL (Mukhoti et al., 2023) | 13.2 | 21.0 | 16.0 | 60.4 | 12.9 |
| TCL no ensemble (Cha et al., 2023) | 14.1 | 28.7 | 22.0 | 76.7 | 18.6 |
| TCL (Cha et al., 2023) | 14.9 | 30.3 | 23.1 | **77.5** | 19.6 |
| **SILC\* (Ours)** | 17.2 | 29.3 | 25.1 | 73.5 | 18.2 |
| **SILC (Ours)** | **19.3** | **31.6** | **26.9** | **77.5** | **20.8** |

Table 4: **Additional Zero-shot Semantic Segmentation comparisons.** We report additional results for the previous state-of-the-art TCL. We additionally report result for another baseline PACL. SILC consistently outperforms the baselines on the same evaluation protocol i.e. raw predictions of the model.

image-text pairs for this experiment and report the performance in Table 4. We observe that PACL performs worse than TCL and SILC . Since our reproduced numbers are different from the reported numbers in PACL manuscript, we contacted the authors and discussed their evaluation protocol in detail. PACL uses segmentation label supervision at test time and tunes a threshold on the model's prediction to only extract image regions where the model has a high confidence. The segmentation performance is only evaluated over these regions and not the full label from the dataset. Since we are interested in the raw zero-shot semantic segmentation performance of the model over the full image, we do not perform this step and show that SILC outperforms PACL.

# 6 ADDITIONAL QUALITATIVE RESULTS.

## 6.1 ADDITIONAL QUALITATIVES ON ZERO-SHOT SEMANTIC SEGMENTATION.

We report additional qualitative results for Zero-Shot Semantic Segmentation in Figure 1 for A-150 and Figure 2 for PC-59. They demonstrate that SILC produces less noisy segmentations compared to CLIP and is less prone to class confusions such as booth/computer, field/grass, road/screen, swivel chair/chair, blind/curtain, counter/countertop, counter/kitchen, rock/mountain, rock/sand, animal/sea, armchair/sofa, and food/glass.

## 6.2 ADDITIONAL QUALITATIVES ON OPEN VOCABULARY SEMANTIC SEGMENTATION.

We report additional qualitative results for Open Vocabulary Semantic Segmentation in Figure 3 for A-150 and Figure 4 for PC-459. They demonstrate that SILC better distinguishes semantically similar classes such as bookcase/shelf, countertop/counter, cabinet/shelf, swivel chair/ chair, stool/chair, pier/bridge, desk/shelf, train/metal, building/shed, wall/brick, sign/poster, cloth/plastic, ground/sand, and boat/water.

# 7 ADDITIONAL DETAILS.

## 7.1 EVALUATION PROTOCOL.

We follow the original CLIP (Radford et al., 2021) paper for the zero-shot classification and retrieval evaluations. We follow the original ViT (Dosovitskiy et al., 2021) paper for few-shot classification

| Image | CLIP | SILC | G. Truth | Image | CLIP | SILC | G. Truth |

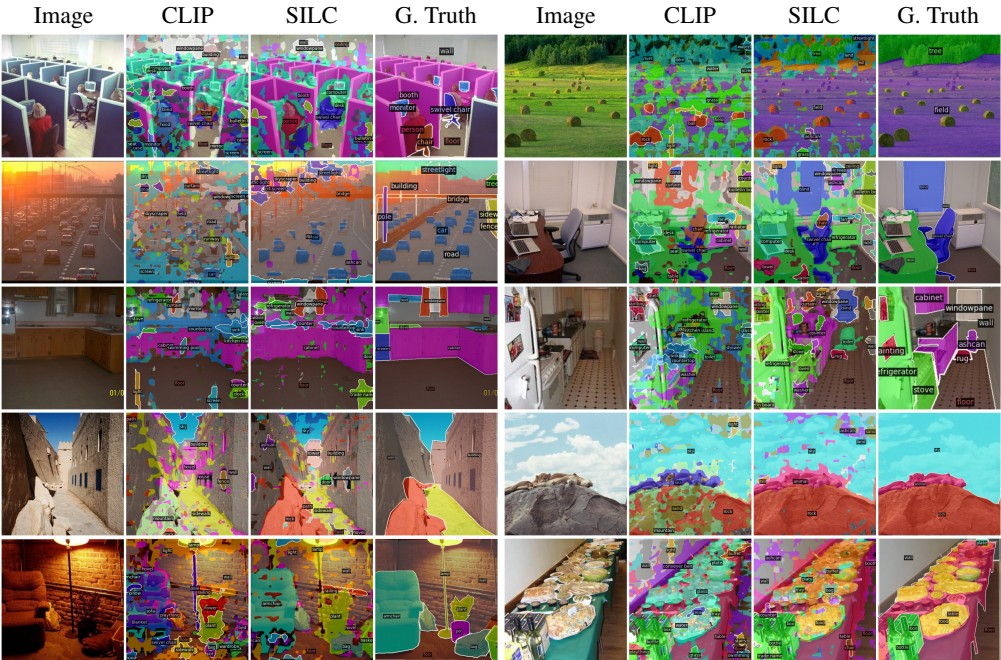

Figure 1: **Additional qualitative results for zero-shot segmentation on A-150.**

evaluation. The evaluation code is used from the `big_vision` codebase (Beyer et al., 2022a;b). For our segmentation evaluations, we export our model weights to PyTorch. We follow previous works (Cha et al., 2023; Xu et al., 2022; Shin et al., 2022; Zhou et al., 2022) and implement our zero-shot segmentation evaluation in MMSeg (Contributors, 2020) with Sliding-Window evaluation. We directly use the model's prediction for segmentation and do not perform any refinement. For Open Vocabulary segmentation, we directly use the codebase from Cat-Seg (Cho et al., 2023) and do not perform any hyper-parameter tuning. All results for Cat-Seg are reported using the training protocol from the authors.

## 7.2 Additional Training Details.

We provide training detail for SILC* and SILC in the main manuscript. We provide additional training detail in this supplementary. SILC* at ViT B/16 can be trained on 256 TPUv4 chips meanwhile the B/8 and L/16 models require 512 chips. Our model requires more compute compared to CLIP as we keep an EMA copy of model weights as well as do more forward passes for the ViT. However we show in our main manuscript Section 4.6 that SILC* already outperforms CLIP at 1/4th of example-seen. For the fine-tuning stage for SILC , we use a initial learning rate of $1e^{-4}$ and use a rsqrt scheduler Zhai et al. (2022) with 50000 cool down steps. We do not use warm up or weight decay at this stage. The MLP used for our self-distillation loss consists of two layers with gelu activation and dimension of 2048. This is followed by a bottleneck of dimension 256 followed by a projection to the output dimension $K$ of size 65536. We do not perform tuning of each loss's contribution and directly optimise the sum of loss coming from our model's two components.

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

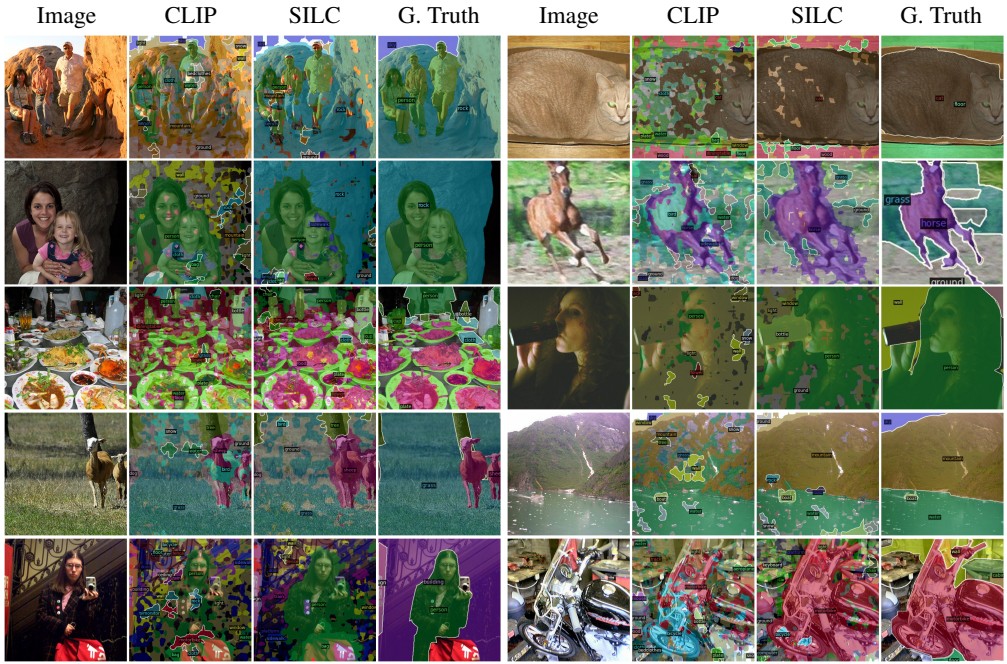

Figure 2: **Additional qualitative results for zero-shot segmentation on PC-59.**

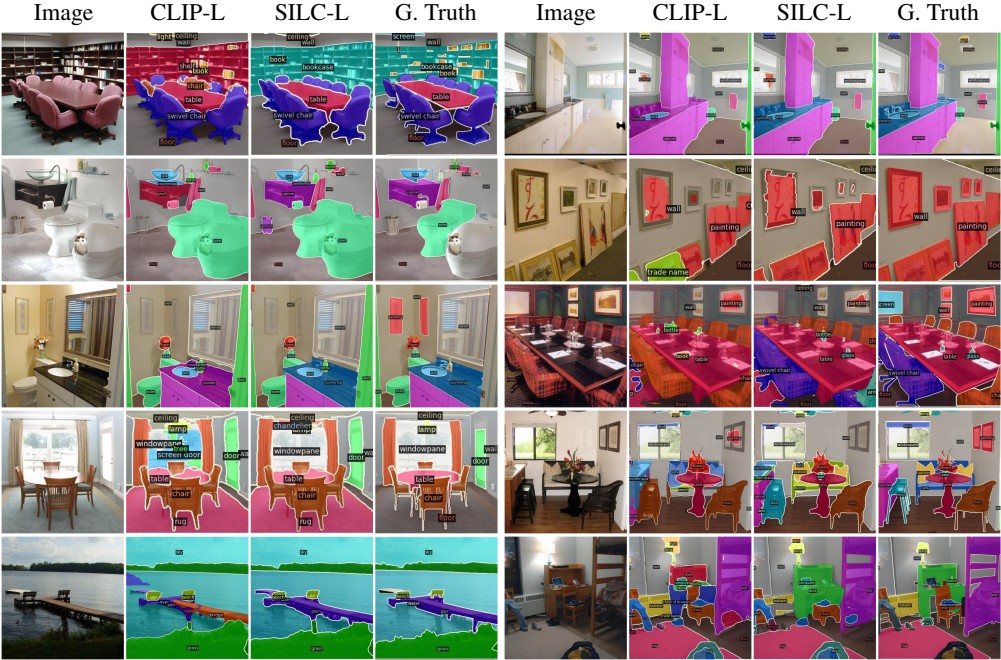

Figure 3: **Additional qualitative results for open-vocabulary segmentation on A-150.**

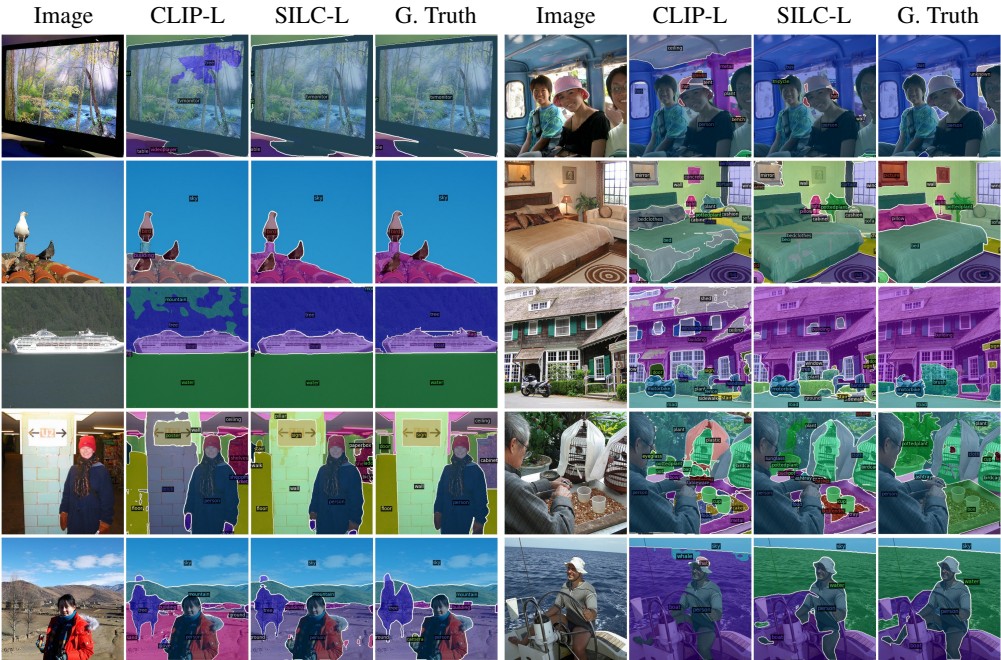

Figure 4: **Additional qualitative results for open-vocabulary segmentation on PC-459.**

Junbum Cha, Jonghwan Mun, and Byungseok Roh. Learning to generate text-grounded mask for open-world semantic segmentation from only image-text pairs. In *CVPR*, 2023.

Seokju Cho, Heeseong Shin, Sunghwan Hong, Seungjun An, Seungjun Lee, Anurag Arnab, Paul Hongsuck Seo, and Seungryong Kim. Cat-seg: Cost aggregation for open-vocabulary semantic segmentation. *arXiv preprint arXiv:2303.11797*, 2023.

MMSegmentation Contributors. MMSegmentation: Openmmlab semantic segmentation toolbox and benchmark. https://github.com/open-mmlab/mmsegmentation, 2020.

Alexey Dosovitskiy, Lucas Beyer, Alexander Kolesnikov, Dirk Weissenborn, Xiaohua Zhai, Thomas Unterthiner, Mostafa Dehghani, Matthias Minderer, Georg Heigold, Sylvain Gelly, Jakob Uszkoreit, and Neil Houlsby. An image is worth 16x16 words: Transformers for image recognition at scale. In *ICLR*, 2021.

Jishnu Mukhoti, Tsung-Yu Lin, Omid Poursaeed, Rui Wang, Ashish Shah, Philip HS Torr, and Ser-Nam Lim. Open vocabulary semantic segmentation with patch aligned contrastive learning. In *Proceedings of the IEEE/CVF Conference on Computer Vision and Pattern Recognition*, pp. 19413–19423, 2023.

Maxime Oquab, Timothée Darcet, Théo Moutakanni, Huy Vo, Marc Szafraniec, Vasil Khalidov, Pierre Fernandez, Daniel Haziza, Francisco Massa, Alaaeldin El-Nouby, et al. Dinov2: Learning robust visual features without supervision. *arXiv preprint arXiv:2304.07193*, 2023.

Alec Radford, Jong Wook Kim, Chris Hallacy, Aditya Ramesh, Gabriel Goh, Sandhini Agarwal, Girish Sastry, Amanda Askell, Pamela Mishkin, Jack Clark, et al. Learning transferable visual models from natural language supervision. In *ICLR*, 2021.

Gyungin Shin, Weidi Xie, and Samuel Albanie. Reco: Retrieve and co-segment for zero-shot transfer. In *NeurIPS*, 2022.

Yongqin Xian, Christoph H Lampert, Bernt Schiele, and Zeynep Akata. Zero-shot learning—a comprehensive evaluation of the good, the bad and the ugly. *IEEE transactions on pattern analysis and machine intelligence*, 41(9):2251–2265, 2018.

Jiarui Xu, Shalini De Mello, Sifei Liu, Wonmin Byeon, Thomas Breuel, Jan Kautz, and Xiaolong Wang. Groupvit: Semantic segmentation emerges from text supervision. In *CVPR*, 2022.

Xiaohua Zhai, Alexander Kolesnikov, Neil Houlsby, and Lucas Beyer. Scaling vision transformers. In *CVPR*, 2022.

Xiaohua Zhai, Basil Mustafa, Alexander Kolesnikov, and Lucas Beyer. Sigmoid loss for language image pre-training. In *ICCV*, 2023.

Chong Zhou, Chen Change Loy, and Bo Dai. Extract free dense labels from clip. In *ECCV*, 2022.