# OpenReview forum: "SILC: Improving Vision Language Pretraining with Self-Distillation"
_ICLR.cc/2024/Conference — ICLR 2024 Conference Withdrawn Submission_

### Official Review · Reviewer_FBGc · 2023-10-27

**Soundness:** 2 fair
**Presentation:** 3 good
**Contribution:** 2 fair
**Rating:** 3
**Confidence:** 4

**Summary:**

This paper proposes a pre-training framework for vision-language models, termed SILC, to enhance local image features by self-distillation. The experiments are conducted on several tasks including classification, retrieval and segmentation, and SILC achieves the comparable performance with the baseline methods.

**Strengths:**

This work focuses on improving the performance of vision-language models on dense prediction tasks by designing a pre-training strategy, which is critical in the real-world applications. Experiments on both global recognition and dense prediction tasks are conducted and reported.

**Weaknesses:**

The important technical details, especially about the local crop, should be further clarified and explained.

- The operation details of how to obtain the local crop are important and expected to be clarified, since only applying additional views has introduced significant performance improvement as shown in Table 4.

- The explanation of why the performance is improved by enforcing the embeddings of local crop to be similar to the teacher prediction. As the local view is ``a random crop over a small region of the image", it may only contain the background or part of the foreground.

- The proposed SILC enhances the model local representation by local-to-global correspondence learning, which promotes the applications of VLM on segmentation tasks. However, how does this training strategy improve the performance of not only dense prediction tasks but also global recognition tasks?

More convincing experimental results should be provided.

- Direct comparisons with the results reported by the existing works (e.g., system-level comparisons or results under their settings) are expected. This may avoid concerns about fair comparisons, such as whether the baseline methods are under-tuned when they are reimplemented.

- The results of SILC* should be used for comparison since the setting of SILC is unfair. Compared with the baseline methods, the improvements of SILC* on several tasks are marginal.

- In order to demonstrate the effectiveness of the proposed approach on dense prediction tasks, the results for open-vocabulary object detection are critical, which is missing.

**Questions:**

See Weaknesses.

---

> ### Author Response · Authors · 2023-11-17
>
> - Weakness 1: the local crop are important and expected to be clarified
>
> As mentioned in the manuscript we use random cropping without any specific constraints.
>
> - Weakness 2/3 : why the performance is improved by enforcing the embeddings of local crop to be similar to the teacher prediction, how does this training strategy improve the performance of not only dense prediction tasks but also global recognition tasks.
>
> Local-to-global consistency learning allows the model to learn features which are more locally aware. This allows for learning stronger vision backbones and therefore positively impacts all tasks as also shown in several SSL works.
>
> - Weakness 4: Direct comparisons with the results reported by the existing works
>
> It is not possible to directly compare with author results of respective works in Table 1 as they have only trained for small training durations on smaller datasets. Their numbers as a result are much weaker. We are the first work that scales every baseline under the same data to measure their impact on performance.
>
> - Weakness 5: Compared with the baseline methods, the improvements of SILC* on several tasks are marginal.
>
> Can the reviewer please point out where they think improvements are “marginal”? We have consistent improvements on 5 different computer vision tasks. e.g.
>
> Zeroshot classification on Imagenet: 74.1 -> 75.3
>
> COCO Retrieval: 61.7 -> 62.5 and 43.9 -> 44.9
>
> Zeroshot Segmentation on ADE-150: 15.0 -> 17.2
>
> Zeroshot Segmentation on PC-59: 24.0 -> 29.3
>
> - Weakness 6: the results for open-vocabulary object detection are critical, which is missing
>
> This will be added in our future submission.

---

### Official Review · Reviewer_XE72 · 2023-10-28

**Soundness:** 3 good
**Presentation:** 3 good
**Contribution:** 3 good
**Rating:** 5
**Confidence:** 3

**Summary:**

The paper introduces a method called SILC (Self-Distillation for Improving Vision-Language Pretraining with Contrastive learning) that aims to improve vision-language pretraining models for dense prediction tasks such as segmentation. The authors propose the addition of local-to-global correspondence learning by self-distillation as an additional objective for contrastive pre-training. They show that distilling local image features from an exponential moving average (EMA) teacher model significantly improves model performance on various computer vision tasks, including classification, retrieval, and segmentation. SILC sets a new state of the art for zero-shot classification, few-shot classification, image and text retrieval, zero-shot segmentation, and open vocabulary segmentation.

**Strengths:**

1.	SILC applies local-to-global correspondence learning by self-distillation as an additional objective for contrastive pre-training.
2.	Distilling local image features from an EMA teacher model improves model performance on various computer vision tasks.
3.	SILC sets a new state of the art for zero-shot classification, few-shot classification, image and text retrieval, zero-shot segmentation, and open vocabulary segmentation.

**Weaknesses:**

1.	The novelty of the paper is a little bit concerning, as the local-global consistency training has already been widely used in a number of works (e.g. Caron et al., 2021; Oquab et al., 2023; Zhou et al., 2022b). EMA is also a very common technique widely used in self-/semi- supervised learning community. The limitation is also pointed out by the paper.
2.	SILC claims to improve vision-language pretraining models for “dense prediction tasks”. However, only segmentation related tasks are validated in the experiments. Please also consider adding more experimental comparisons about open-vocabulary object detection (which is also very important topics for dense prediction).
3.	Missing comparisons with some state-of-the-art CLIP variants. Please consider comparing with SOTA CLIP variants, e.g. EVA-02-CLIP[a].
4.	Recently, there is a concurrent work [b] which uses self-distillation for improving dense representation ability of CLIP, which is very related to this work. Please consider adding discussion about it in the related work section.
5.	Please report the training cost (training time) of the model.
6.	Please consider discussing the limitation of the paper.

Minor:
1)	Presentation of Figure1 can be improved. Please consider using different colors for the arrows and losses, to make it easier to understand. Especially, in the figure, the student takes both local crop and global crop as the input. But they are not used at the same process.

[a] Sun, Quan and Fang, Yuxin and Wu, Ledell and Wang, Xinlong and Cao, Yue. EVA-CLIP: Improved Training Techniques for CLIP at Scale. Arxiv2023.
[b] Size Wu and Wenwei Zhang and Lumin Xu and Sheng Jin and Xiangtai Li and Wentao Liu and Chen Change Loy. CLIPSelf: Vision Transformer Distills Itself for Open-Vocabulary Dense Prediction. Arxiv2023.

**Questions:**

Please see above. The paper is well presented. However, the reviewer concerns the somewhat limited novelty, insufficient experiments (especially for open-vocabulary detection).

---

> ### Author Response · Authors · 2023-11-17
>
> - Weakness 1: The novelty of the paper is a little bit concerning
>
> We have introduced our contribution as simple but effective in the manuscript. We have included sufficient attribution and citation to DINO. To the best of our knowledge we are the first work to show that the DINO objective can be complementary to the CLIP objective. DINO has only shown their conclusions for class agnostic segmentation features and is unable to provide any class level inference out of the box. Therefore, we believe that our work has several important practical implications not achievable by the respective DINO works.
>
> - Weakness 2: Please also consider adding more experimental comparisons about open-vocabulary object detection
>
> These will be added in the future version of the manuscript.
>
> - Weakness 3: Please consider comparing with SOTA CLIP variants,  e.g. EVA-02-CLIP[a].
>
> Our best SILC models outperform EVA CLIP (e.g. 76.6 vs  74.7 ZS on Imagenet at ViT/B16). EVA CLIP does not propose a new training algorithm but rather better training strategies that utilize pretrained vision and language encoders separately similar to LiT-Tuning[b]. While EVA CLIP does offer significantly improved performance over OpenAI’s CLIP, we compare with backbones trained on the same data in Table 1 to quantify the impact of the training algorithm.
>
> - Weakness 4: Recently, there is a concurrent work [b] which uses self-distillation for improving dense representation ability of CLIP
>
> The mentioned work is a concurrent submission to ICLR which only came online after our paper submission. It is realistically not possible for us to have cited a work that was submitted to the same double blind review process as ours. SILC models offer superior performance to the mentioned work.
>
> - Weakness 5: Please report the training cost (training time) of the model.
>
> For large scale training, data and compute are both important axes as we compare model performance. While SILC does result in additional training cost with respect to core-hours, we show in Table 5 that SILC at 5B example-seen already outperforms CLIP. This baseline of SILC consumes approximately the same core-hours while having consumed 1/4th of the data. Since foundation models are designed to be trained once and used in multitude of applications(e.g. CLIP models are still used to date 2+ years later), we feel that comparing across the same data to get the best model is the more important axis.
>
>
> Citation:
> [b] LiT: Zero-Shot Transfer with Locked-image text Tuning, CVPR2022

---

### Official Review · Reviewer_tTwz · 2023-11-01

**Soundness:** 2 fair
**Presentation:** 3 good
**Contribution:** 2 fair
**Rating:** 3
**Confidence:** 5

**Summary:**

This study introduces an innovative technique that incorporates self-distillation loss into the pre-training of a vision-language model, aiming to establish local-global consistency. It is achieved through the utilization of an exponential moving average (EMA) teacher model. The proposed approach demonstrates superior scalability compared to the conventional image-text pre-training method and yields enhanced performance in various tasks, including zero-shot classification, few-shot classification, image-to-text and text-to-image retrieval, zero-shot semantic segmentation, and open vocabulary semantic segmentation.

**Strengths:**

1. The method in this paper is simple and has shown to be effective on several downstream tasks including image classification, retrieval, and segmentation.

2. The authors have provided ablation studies on the components of the method and verified the effectiveness of the self-distillation across many downstream tasks.

3. The authors prove better scalability of the method over baseline image-text pre-training on the zero-shot classification on the ImageNet dataset.

**Weaknesses:**

1. The motivation is somewhat unaligned with the experiments. The authors discussed the challenge of open-vocabulary dense prediction tasks including image segmentation and object detection. Conceptually, the proposed method would be established as a solution to the mainstream open-vocabulary dense prediction tasks including semantic segmentation, instance segmentation, and object detection, by imposing the global-local consistency that has been proven effective for these tasks in the literature of Self-supervised Learning. However, the authors have only provided experimental support for the proposed semantic segmentation method and a wide range of image-level tasks.
2. This paper seems to be incremental and lacks novelty. The key observation that segmentation emerges from learning local and global consistency, which sets the ground for the self-distillation approach, is contributed to self-supervised learning works such as DINO. Also, an EMA teacher has been a common practice in the self-supervised learning domain. Therefore, the technical contribution of this paper is quite limited.
3. The authors have attempted to showcase the scalability of the proposed method, but only on the zero-shot classification on the ImageNet dataset. I believe such a claim should undergo a more rigorous validation on more downstream tasks, including the dense prediction ones considered in this paper.
4. The authors missed a closely related work published on CVPR 2023, i.e., PACL [1]. The latter applied a Patch Aligned Contrastive Learning approach to the image-text pretraining to enforce vision-language alignment at the dense level. Discussion and quantitative comparison with PACL is needed, especially on the zero-shot semantic segmentation benchmark. Note that PACL is trained on million-scale data instead of the billion-scale data used by SILC.
5. For the comparison in Table 2, the compared methods are trained with smaller datasets, e.g., GroupViT is trained on CC12M and YFCC(14M). Although we desire a larger-scale training of vision-language foundation models, a fair comparison with existing methods should also be presented.

[1] Open Vocabulary Semantic Segmentation with Patch Aligned Contrastive Learning.  Mukhoti et.al., CVPR2023

**Questions:**

1. The impact of SILC on object detection and instance segmentation needs to be verified.
2. How about CNN-based VLMs? There are recent works [2, 3] that reveal the effectiveness of CNN-based CLIP models for open-vocabulary detection and segmentation.  The frozen CLIP CNNs without any additional losses or finetuning can already achieve SOTA on open-vocabulary object detection and panoptic segmentation. Would this approach also apply to CNN-based models?

[2] F-VLM: Open-Vocabulary Object Detection upon Frozen Vision and Language Models. Kuo et.al., ICLR2023.

[3] Convolutions Die Hard: Open-Vocabulary Segmentation with Single Frozen Convolutional CLIP. Yu et.al., NeurIPS2023.

---

> ### Author Response · Authors · 2023-11-17
>
> - Weakness 1: authors have only provided experimental support for the proposed semantic segmentation method and a wide range of image-level tasks.
>
> We have evaluated SILC for further downstream tasks and found consistent improvements. This will be included in the next version of the manuscript at a future venue.
>
> - Weakness 2:  paper seems to be incremental
>
> We have introduced our contribution as simple but effective in the manuscript. We have included sufficient attribution and citation to DINO. To the best of our knowledge we are the first work to show that the DINO objective can be complementary to the CLIP objective. DINO has only shown their conclusions for class agnostic segmentation features and is unable to provide any class level inference out of the box. Therefore, we believe that our work has several important practical implications not achievable by the respective DINO works.
>
> - Weakness 3: authors have attempted to showcase the scalability of the proposed method, but only on the zero-shot classification on the ImageNet dataset.
>
> We had evaluated the performance of scaling across all metrics but only included the zero-shot classification numbers as they tend to be the most common metric. The conclusion on other metrics remains the same. We thank the reviewer for the helpful suggestion and will include further analysis in the future version of the manuscript.
>
> - Weakness 4: The authors missed a closely related work published on CVPR 2023, i.e., PACL [1]
>
> We thank the reviewer for pointing this out. We have already discussed PACL in detail in our supplementary. We agree with the reviewer that a reference to this should be mentioned in the main manuscript. We will include a reference to this result in the main manuscript in our future submission so it's easier to find.
>
> We reproduced PACL on the same WebLI dataset and found it to achieve inferior performance to SILC while having significant extra memory cost due to patch-wise attention. Patchwise attention matrix prevents PACL from scaling to large batch sizes which is especially important for contrastive learning.
>
> Moreover, the PACL manuscript uses a non-standard evaluation protocol where it only evaluates the segmentation performance on high confidence prediction of the model instead of over the full image and dataset. Therefore their results are not directly compare with other methods including ours which evaluates over the full dataset. Please refer to our supplementary for detailed discussion.
>
> - Weakness 5: GroupViT is trained on CC12M and YFCC(14M). Although we desire a larger-scale training of vision-language foundation models, a fair comparison with existing methods should also be presented.
>
> GroupViT has been shown to be limited in its performance beyond object classes due to architectural design[a]. We have compared patch alignment on frozen VLM in our supplementary on WebLI data. We found it to offer inferior performance compared to the SILC objective.
>
> TCL and ReCo are designed to improve the zero-shot segmentation result of a frozen VLM. These two baselines can potentially offer additional performance if used with SILC backbones which offer better zero-shot segmentation performance to start from. However, our objective is to build a single foundation model that can improve all computer vision tasks. These baselines are only specialized to zero-shot segmentation. Therefore it is out of scope of our study.
>
>
> Citation:
> [a] Learning to Generate Text-grounded Mask for Open-world Semantic Segmentation from Only Image-Text Pairs. CVPR2023

---

### Official Review · Reviewer_K27w · 2023-11-02

**Soundness:** 3 good
**Presentation:** 3 good
**Contribution:** 2 fair
**Rating:** 5
**Confidence:** 4

**Summary:**

This paper proposed a pre-training method, SILC, to improve vision-language tasks, especially zero-shot/open vocabulary semantic segmentation tasks. The method is a combination of CLIP and self-distillation methods (e.g. DINO), which enhance the local features of CLIP models for better performance on dense prediction tasks.

**Strengths:**

1. The method is simple and effective. Although it does not introduce novel objectives for pre-training, it shows a simple combination of CLIP and DINO can work well at a relatively large scale.
2. The presentation is clear and easy to follow.

**Weaknesses:**

1. The experiments are not convincing. In Table 1, the authors reproduce several baseline methods for comparison under a comparable setting. However, in Table 2, only the reproduced CLIP has been compared. In Table 3, only open-sourced CLIP has been compared. The adaptation costs of those reproduced methods are relatively small compared to the pre-training costs. I think the authors should also compare with those baseline methods in Table 2 and 3.
2. For large-scale pre-training, computation efficiency is also important. The proposed methods use 2 global views and 8 local views for pre-training, which should be much slower in practice. Using "example-seen" is not a good indicator. I think the authors should compare the computation costs with the referred baseline methods.

**Questions:**

See the weaknesses part.

---

> ### Author Response · Authors · 2023-11-17
>
> - Weakness 1: Table 2 and 3 should be extended with additional baselines of Table 1.
>
> We thank the reviewer for their interest in additional results. These baselines had similar results to CLIP(WebLI) so we excluded them due to space constraints. Results of OV segmentation (Table 3) for additional CLIP(WebLI) and SILC* are already available in the supplementary.
>
> - Weakness 2: computation efficiency is also important, The proposed methods use 2 global views and 8 local views for pre-training, which should be much slower in practice.
>
> We thank the reviewer for raising this important point. We want to make a slight correction, we use 6 local views not 8. For large scale training, data and compute are both important axes as we compare model performance. While SILC does result in additional training cost with respect to core-hours, we show in Table 5 that SILC at 5B example-seen already outperforms CLIP. This baseline of SILC consumes approximately the same core-hours while having consumed 1/4th of the data. Since foundation models are designed to be trained once and used in multitude of applications(e.g. CLIP models are still used to date 2+ years later), we feel that comparing across the same data to get the best model is the more important axis.

---

### Author Response · Authors · 2023-11-17

We thank the reviewers for their feedback on work. Reviewers found our contribution to be “simple and effective”, “simple and has shown to be effective on several downstream tasks”, “improves model performance on various computer vision tasks” and “critical in the real-world applications”.

Since this submission, we have made several improvements to SILC including extending it to SigLIP objective and extending its application to even more computer vision tasks including open vocabulary detection, captioning and VQA. SILC achieves consistently improvements on all these additional tasks. We feel these additions require a resubmission to a future venue. We therefore thank the reviewers again for their hard work and will incorporate their feedback in the new version of the work. We answer individual reviewers' concerns below.